# Dehydration Stress Memory Genes in *Triticum turgidum* L. ssp. *durum* (Desf.)

**DOI:** 10.3390/biotech11030043

**Published:** 2022-09-13

**Authors:** Monther T. Sadder, Anas Musallam, Majd Allouzi, Mahmud A. Duwayri

**Affiliations:** 1Plant Biotechnology Lab, Department of Horticulture and Crop Science, School of Agriculture, University of Jordan, Amman 11942, Jordan; 2National Agricultural Research Center, Amman 19381, Jordan

**Keywords:** abiotic, drought, epigenetics, physiology, transcriptional memory, wheat

## Abstract

Exposure to successive stress cycles can result in a variety of memory response patterns in several plant species. We have investigated a group of these patterns at both the transcriptional and physiological memory levels in durum wheat. The data revealed huge discrepancies between investigated durum wheat cultivars, which presumably are all drought tolerant. It was possible to generate a consensus memory response pattern for each cultivar, where Hourani 27 was the most tolerant followed by Balikh 2 and then Omrabi 5. When durum wheat homologs from rice and maize were compared, only 18% gave similar memory response patterns. The data would indicate the presence of potentially divergent memory mechanisms in different plant species and genotypes. Ultimately, a thorough examination is required for each genotype before giving solid memory-based conclusions that can be applied in plant breeding and agricultural management practices.

## 1. Introduction

Durum wheat is a major food industry product, which is used mainly for making pasta, semolina and bulgur. Currently available durum wheat landraces and modern cultivars (*Triticum turgidum* L. ssp. *durum* (Desf.) Husn.) are the product of long-term domestication from wild emmer wheat (*T. turgidum* ssp. *dicoccoides* (Körn. ex Asch. and Graebn.) Thell. Jordan is considered a major center of origin for wheat, where immense genetic diversity between local durum wheat genotypes is evident [1,2,3]. Such diversity is reflected in major yield components (biological yield, fertile tillers, number of seeds per spike and seed weight) and adaptation to diverse abiotic stresses, e.g., drought, salt, heat and cold. Characterization of physiological and phenological parameters of such genetic resources is crucial for wheat ideotype development for changing environments [4,5,6]. Therefore, research was focused on valorization of these invaluable genetic resources to generate novel breeding lines with improved traits [7,8,9].

Transcriptional changes can lead to successful adaptation and eventually tolerance in different plant species. However, if plants fail to adapt to a stressful environment, they are considered sensitive to that condition. Therefore, expression profiling can define both tolerant and sensitive plant responses [10]. These profiles can uncover specific regulators necessary to elevate stress tolerance and can be used as tools to study regulatory genes [11]. Such variation is evident between modern and ancient varieties, e.g., durum wheat modern varieties producing significantly higher yield and showing better nitrogen use efficiency compared to their ancient counterparts [12]. On the contrary, ancient varieties can mitigate N and water stresses much better than modern varieties. Transcriptional regulation of tolerant phenotypes is controlled by several responsive genes but mainly through DNA-binding transcription factors (TFs) [13,14]. Moreover, plants that have been pre-exposed to a stress (priming) may produce altered cellular, biochemical, and/or transcriptional responses to a subsequent stress of a similar nature. This behavior could benefit the plant by making it more resistant to future stresses, leading to the idea that plants exercise a form of ‘memory’ from the previous stress [15,16]. Using a genome-wide RNA-Seq approach, the transcriptional responses in Arabidopsis plants, which have experienced multiple exposures to dehydration stress were investigated [17,18]. The data revealed four distinct, previously unknown, transcription ‘memory’ response patterns of dehydration stress genes. A related study on maize showed 2062 genes, which have dehydration response patterns similar to Arabidopsis [19]. Furthermore, 6885 rice transcripts and 238 lncRNAs were found to have a sort of memory under cycles of drought stress, which were grouped into 16 unique response patterns [20]. More recently, related drought memory genes were also revealed in potato, which include a diverse array of genes involved in photosynthesis, signal transduction, sugar metabolism, protease and protease inhibitors, flavonoids metabolism, transporters and TFs [14]. However, data are almost absent for memory drought-responsive genes in durum wheat. Therefore, this study aims at investigating the response pattern of drought genes in durum wheat, where three different genotypes were assessed to elucidate genotype interactions, if any, with memory genes.

## 2. Materials and Methods

### 2.1. Plant Materials

Three durum wheat cultivars (*Triticum turgidum* L. subsp. *durum*) were used in this study; Hourani 27, Omrabi 5 and Balikh 2 (National Agricultural Research Center, Amman, Jordan). Hourani 27 is a Jordanian improved selection from the Hourani landrace and is considered a drought-tolerant cultivar [21]. The Omrabi 5 (syn. Om Qais) cultivar is a cross between the landrace Hourani and the improved cultivar “Jori-C69”, which was also proved to be drought tolerant [22], while Belikh 2 was bred at ICARDA (Crane/Stork) and developed for rain-fed areas; it is also early heading and maturing cultivar with good protein quality for pasta processing. Both Omrabi 5 and Belikh 2 were developed for the Mediterranean dryland conditions [23].

### 2.2. Dehydration Stress

Wheat seeds were germinated over moist filter paper inside polypropylene vessels (500 mL). Two-week-old seedlings were subjected to stress treatment following a modified procedure as previously described [19]. Seedlings were removed from vessels, and roots were gently blotted onto filter paper to remove excess water and then air-dried for 2 h (first dehydration stress, S1). This step was followed by a period of re-hydration recovery for 24 h. An additional stress cycle was performed as described above (second dehydration stress, S2). For control watered plants (W), seedlings were not exposed to any dehydration stress.

### 2.3. Physiological Measurements

Photosynthesis (PSII) activity was determined for leaves of W, S1 and S2 seedlings by measuring transient chlorophyll fluorescence using Handy PEA (Hansatech, King’s Lynn, UK) with an excitation light energy of 3000 μmol m^−1^ s^−1^. Moreover, stomatal conductance (gs) was measured using SC-1 porometer (Decagon, Pullman, WA, USA). Leaf relative water content (RWC) was measured as described [24] with minor modifications, where young leaves (7–9 cm long) were immersed in 3 mL of deionized water in 15 mL secured conical tubes for 24 h at RT in the dark. Means were separated with LSD (*p* < 0.5%). The data were further presented as relative measure as compared to control plants. Finally, pair-wise relationships between physiological parameters were plotted for all measured physiological parameters.

### 2.4. Analysis of Dehydration Responsive Memory Genes Using Quantitative Real-Time PCR

At the end of each dehydration treatment, leaves were collected, flash frozen in liquid nitrogen and subsequently stored at −80 °C. Total RNA was extracted using guanidinium thiocyanate-phenol-chloroform procedure [25] with TRIsure (Bioline, Memphis, TN, USA). RNA was dissolved in DEPC-treated water supplemented with RNase inhibitor (Qiagen, Hilden, Germany). Reverse transcription reactions were performed following manufacture procedure using GoScript™ Reverse Transcriptase kit (Promega, Madison, WI, USA). Memory, non-memory and late-response genes for durum wheat were retrieved from durum wheat annotated genome based on maize orthologs [19] (Table 1). Specific primers were designed for twelve wheat dehydration stress memory responsive genes (Table 1) and utilized to assess relative gene expression using quantitative real-time polymerase chain reaction (qPCR) as described earlier using 2^ΔΔCT^ method and 95% confidence interval (95% CI, z score = 1.96) [26].

## 3. Results

Two successive dehydration stresses interrupted by short recovery period were applied in this study to assess both physiological and molecular memory responses in durum wheat. RWC showed a drop after the first stress (S1). An additional dramatic drop was recorded after the second dehydration stress (S2) (Figure 1A). Although this decline was evident in all three investigated durum wheat cultivars, Balikh 2 showed the highest drop in relative RWC value, which was generated by comparing RWC data under stress to the control (W), and recorded a 1.22-fold decline as compared to the control (Figure 2). Likewise, stomatal conductance showed a subsequent stepwise decline during S1 and S2 stress treatments for all durum wheat cultivars (Figure 1B). It is interesting to note that different durum wheat cultivars showed a huge variation in stomatal conductance even in unstressed seedlings (W), where Balikh 2 had the lowest control value and a record 1.8-fold decline in stomatal conductance in S2 as compared to its counterpart control (Figure 2). On the other hand, PSII activity was barely affected during S1 and S2 dehydration stresses in Hourani 27 and Omrabi 5, while it decreased sharply during S2 in Balikh 2 cultivar (Figure 1C). Pairwise correlation coefficients were calculated for the investigated parameters (PSII activity, stomatal conductance and RWC). Correlations (data not shown) were weak and insignificant, except between RWC and stomatal conductance with a moderate correlation (R^2^ = 54%), which was significant at *p* < 0.05.

Plotting the distribution of RWC in durum wheat leaves against stomatal conductance (Figure 3A–C) revealed that control seedlings (W) of Hourani 27 cultivar have a relatively high stomatal conductance and hence the lowest RWC (Figure 3A). Nonetheless, Hourani 27 shifted to the highest RWC level in S1 (Figure 3B), while it regained its position back again in S2, but this time with a relatively lower stomatal conductance and RWC (Figure 3C). On the other hand, Omrabi 5 had the highest stomatal conductance with relatively high RWC during S1 (Figure 3C). The increase in PSII in Omrabi 5 during all treatments (W, S1 and S2) was associated with more gas exchange as indicated by a relatively high stomatal conductance when compared to the other two cultivars (Figure 3D–F); on the contrary, Balikh 2 maintained the lowest stomatal conductance with intermediate PSII activities along all treatments (W, S1 and S2). The comparison of RWC under control conditions with PSTII (Figure 3G), reveals that seedlings of Omrabi 5 have the highest value. Following the second dehydration stress (S2), Omrabi 5 consistently showed higher RWC and higher PSII. The reverse situation (lower RWC and lower PSII) was observed in seedlings of Hourani 27 and Balikh 2 (Figure 3I).

To build a holistic view of dehydration stress memory response in durum wheat, the expression of representative responsive genes was assessed. The genes *TRITD4Bv1G010710* coding for lipoxygenase 2 (Figure 4A) and *TRITD3Bv1G183490* coding for protein kinase C-like zinc finger (Figure 4B) would presumably have a positive memory response for maize orthologs (+/+) under repeated dehydration stress (Table 2). This was true for both genes for the Hourani 27 durum wheat cultivar. However, they showed a (−/+) response in Omrabi 5 and the opposite response (+/−) in Balikh 2 durum wheat cultivar. While, the rice orthologs showed (+/−) and (+/+) patterns, respectively.

On the other hand, the durum wheat gene with presumably negative dehydration stress memory response (−/−) as revealed in maize, the *TRITD5Bv1G217630* and coding for basic helix-loop-helix (bHLH) DNA-binding, showed a rather late-response gene behavior in Hourani 27 (Figure 4C), while it gave a non-memory repression response (−/=) in the other two cultivars. While it showed an opposite (+/+) pattern in rice.

The maize gene *GRMZM2G429322* coding transmembrane amino acid transporter has (+/−) dehydration stress response, which also has a similar pattern in rice (Table 2). Its durum wheat ortholog (*TRITD7Bv1G120550*), gave a (+/+) response in Hourani 27 and a non-memory repression response (−/=) in the other two cultivars (Figure 4D). Another classified (+/−) responsive gene coding for late embryogenesis abundant (LEA), namely, the *TRITD1Av1G156270*, showed a late response (=/+) in rice and Hourani 27 durum wheat, while it showed a (+/−) pattern in the other two cultivars (Figure 4E).

On the other hand, the response pattern (−/+) for the maize gene *GRMZM2G010920* coding for MYB transcription factor, showed a coherent response profile (−/+) in Hourani 27 and Omrabi 5 and a novel response profile (−/−) for Balikh 2 cultivar (Figure 4F) in the durum wheat ortholog *TRITD6Bv1G045800*. The rice ortholog, however, showed the opposite pattern (+/−).

Two non-memory genes were investigated covering both the induced state (+/=) and the repressed state (−/=), based on maize homologs, *GRMZM2G153333* coding for scarecrow-like protein (SCL1) and *GRMZM2G401308* coding for like/winged-helix DNA-binding, respectively (Table 1). The durum wheat ortholog of the former gene, *TRITD3Av1G236010*, showed a memory response of (−/+) in both Hourani 27 and Omrabi 5 cultivars, while it has the opposite response (+/−) response in Balikh 2 (Figure 4G). The durum wheat ortholog of the later gene, *TRITD5Bv1G218230*, showed memory responses of (+/+), (−/+) and (−/−) for Hourani 27, Omrabi 5 and Balikh 2 cultivars, respectively (Figure 4H). On the other hand, the two rice orthologs gave a (−/+) pattern.

Another two genes coding for major plant stress transcription factors with the late-response profile (=/+) were investigated, and again based on maize orthologs (Table 2). They were the *TRITD7Bv1G194910* coding for NAC transcription factor and *TRITD3Bv1G171000* coding for WRKY transcription factor. Interestingly, both genes showed coherent profiles (=/+) in Hourani 27 durum wheat cultivar (Figure 4I,J), respectively. On the contrary, both gave either a (−/+) or (−/−) response in Omrabi 5 and Balikh 2 cultivars, respectively. In rice, the first gave a different pattern of (+/−), while the latter gave a similar pattern (=/+).

The final gene couple of genes were selected to cover the opposite late-response profile (=/−). The durum wheat homologs were *TRITD1Bv1G215920* coding for major facilitator superfamily and *TRITD5Av1G178480* coding for AP2 transcription factor. Both genes showed the opposite extreme profile of a late-response, namely, (=/+) in Hourani 27 durum wheat cultivar (Figure 4K,L), respectively. However, *TRITD1Bv1G215920* showed a memory response (−/+) in Omrabi 5 and Balikh 2 cultivars (Figure 4K). On the other hand, *TRITD5Av1G178480* gave either a (+/−) or (−/−) response in Omrabi 5 and Balikh 2 cultivars, respectively (Figure 4L). In rice, the first gave a different pattern of (+/−), while the latter gave a similar pattern (=/−).

## 4. Discussion

Memory response has long been described in plants subjected to an abiotic stress (or being primed), where expression of key transcription factors is sustained or an epigenetic mark causes a permissive state of expression [27]. It was also described in bread wheat, where drought priming at vegetative growth stages improves tolerance during grain filling [28]. However, the investigated group of major genes showed huge variations in memory response patterns within and between different durum wheat cultivars (Figure 4). Some of these genes were basically regulated through miRNA, e.g., the MYB, which is an important ABA signaling pathways transcription factor in durum wheat [29]. Such small noncoding RNA families were found to be crucial to establish a stress memory feedback loops in several plant species [30]. Therefore, recorded genotype variation in miRNA expression could also influence the memory response in repeated stress cycles. Subsequently, a transcriptional memory would translate, even partially, to a sort of physiological memory, which was demonstrated for maize [31] and correlates well with our findings in durum wheat.

To gain a holistic view of the different patterns of memory genes, hierarchal clustering was performed for each individual durum wheat cultivar (Figure 5). The assessed genes showed unique expression-based clustering for each cultivar, indicating the influence of genotype composition on gene response. The investigated genes showed a consensus pattern of (=/+) in Hourani 27, but not the other two durum wheat cultivars (Figure 5), where Omrabi 5 showed a weak consensus pattern of (−/+), while Balikh 2 showed the stepwise down-regulation (−/−) pattern. Hourani 27 consensus gene behavior (=/+) would indicate a stronger priming effect as compared to the two other cultivars. On the other hand, the consensus pattern (−/+) in Omrabi 5 durum wheat would enable this genotype to withstand successive dehydration stress as compared to the (−/−) consensus pattern in Balikh 2. A field study based on several agronomic traits revealed that Omrabi 5 is actually more drought tolerant than Belikh 2 [32], which agrees with the consensus patterns above. The consensus physiological memory behavior for all three cultivars would give a (−/−) pattern for both RWC and stomatal conductance (Figure 2). On the other hand, the consensus physiological memory behavior showed a more (−/=) pattern for PSTII in durum wheat. Therefore, and with the exception of Balikh 2 cultivar, our data showed a limited collinearity between physiological and molecular memory patterns (Table 2). Nonetheless, they may show more similar patterns for other physiological parameters not investigated in this study or for an additional group of genes.

Furthermore, it is important to highlight the technical dimension for measuring physiological parameters under stress, where some instrumentations may interact with the experimental conditions. For example, the SPAD meter estimates of wheat genotypic variation were found to be satisfactory under warm and irrigated conditions; however, it was less reliable under cooler conditions [33] and varies based on different leaf characteristics [34].

Huge discrepancies between different durum wheat genotypes would indicate a differential epigenetic regulation. In fact, the epigenetics have a profound effect on different plant abiotic stresses [15,16]. In Arabidopsis, an epigenetic-based long-term memory was achieved after osomo-priming [35], while a comparable epigenetics-modulated water stress memory was recorded in barley roots and leaves [36]. A classical example is the Arabidopsis CLF, a member of the Polycomb group, and its product H3K27me3 (histone H3 tri-methylations of Lys 4), which restricted maximal possible induction of dehydration stress-responsive genes but without fully repressing them [37]. Likewise, transcriptional memory (using a transient transcriptional stimulus) was found to stably switch Polycomb target genes in mammals [38]. Moreover, more than three thousand rice memory genes were found to be significantly associated with another epigenetic mark, namely, the CHH DNA methylation [20].

To gain a comparative view using additional genera, expression pattern similarities among investigated durum wheat memory genes were compared with their homologs from maize and rice (Figure 6). Surprisingly, only 18% (two genes) showed similar patterns in all three cereal crops; protein kinase C-like zinc finger with super induced (+/+) pattern and WRKY transcription factor with late-response (=/+) pattern (Table 2). Protein kinase C-like zinc finger was found to be expressed under drought stress in soybeans [39], in addition to salicylic acid [40] and high-light [41] inducible in Arabidopsis. Furthermore, WRKY TFs are known to play a vital role in mitigating drought stress damage in plants including wheat [42]. Likewise, a limited degree of similarity (13.8% on average) was recorded between expression patterns of memory genes between Arabidopsis and their homologs in maize, e.g., 11.9, 8, 15.3 and 8.8% for (+/+), (−/−), (+/−) and (−/+) patterns, respectively [19]. These few ‘universal’ memory factors with common patterns seem to be highly essential for plant survival across multiple genera. Nonetheless, each cereal crop has also some genes with unique expression patterns, e.g., 2, 3 and 4 genes in durum wheat, rice and maize, respectively (Figure 6). This could partially explain the distinctive adaptation behavior for each crop under repetitive drought stress as reflected by memory-responsive genes. Moreover, it is important to highlight the diverse epigenetic marks between distinct individuals (in our case, durum wheat cultivars) as revealed by unique patterns of differentially expression memory genes (Table 2). Therefore, single genotype studies in maize [19] and rice [20] or even Arabidopsis [18] may not reflect the available diverse epigenetic marks related to memory genes. Evidence for global epigenetic diversity was recorded in several plant species under the same genus and even the same species level [43,44,45].

To engage the spatial and temporal expression patterns in the overall view, publicly available data were retrieved for homologs from bread wheat (Figure 7). Two gene homologs were available: *TraesCS4B02G037900* (bread wheat homolog of the durum wheat *TRITD6Bv1G045800* coding for Lipoxygenase 2) and *TraesCS3B02G324400* (bread wheat homolog of the durum wheat *TRITD3Bv1G171000* coding for WRKY transcription factor). The Lipoxygenase 2 encoding gene was up-regulated in different tissues but mainly in pericarp and coleoptile and during both seed germination and seed filling developmental stages. While WRKY transcription factor encoding gene of bread wheat homolog was expressed mainly in coleoptile, radicle tip and root system also during seed germination and seed filling stages. Indicating major developmental and regulatory tasks associated with drought memory responsive genes in wheat.

It is important to pinpoint the importance of stress-specific biomarkers, e.g., heat and drought of salinity [13], which are a reflection of epigenetic modulations that are intern mediated by stress-specific memory genes. This was found to be the case when investigating Arabidopsis under diverse stress conditions, e.g., drought [17,18], heat [46] or cold stress [47].

## 5. Conclusions

Dramatic changes in gene expression were evident following a short-term drought stress in durum wheat. These changes were not strongly reflected in terms of physiological parameters after the first stress. However, they were more prominent after the second stress. It is clear that epigenetic changes in terms of expression of memory genes are fast enough to prime the stressed plant for further stress events, while the physiological response would take more time in durum wheat (five-month life cycle span) than that recorded for Arabidopsis (one-month life cycle span) [18]. Moreover, a major outcome covers differential gene expression patterns between different durum wheat cultivars, which delivers strong evidence supporting unique epigenetic behavior for each specific genotype. Collectively, physiological and molecular memory responsive patterns, when combined with additional vital phenotyping databases, are expected to aid in identifying and developing novel wheat ideotypes for adverse climate change conditions [6,48].

## Figures and Tables

**Figure 1 biotech-11-00043-f001:**
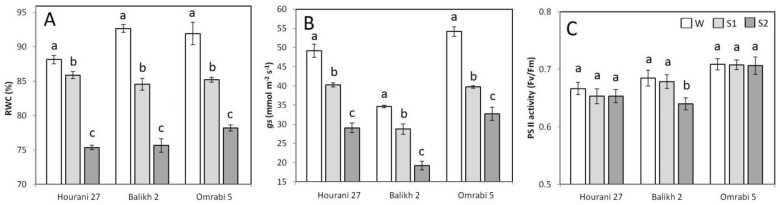
Comparison of RWC (**A**), stomatal conductance (**B**) and PSII activity (**C**) in three durum wheat cultivars (Hourani 27, Balikh 2 and Omrabi 5). Data represent means ± SD. Lower case letters indicate differences between treatments (W, S1 and S2) for each genotype as determined by LSD (*p* < 0.05).

**Figure 2 biotech-11-00043-f002:**
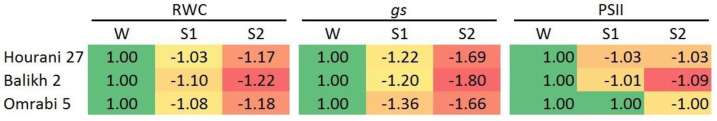
Heat map of relative change in RWC, stomatal conductance (*gs*) and PSII activity in three durum wheat cultivars (Hourani 27, Balikh 2 and Omrabi 5) during S1 and S2 stress stages as compared to the control W.

**Figure 3 biotech-11-00043-f003:**
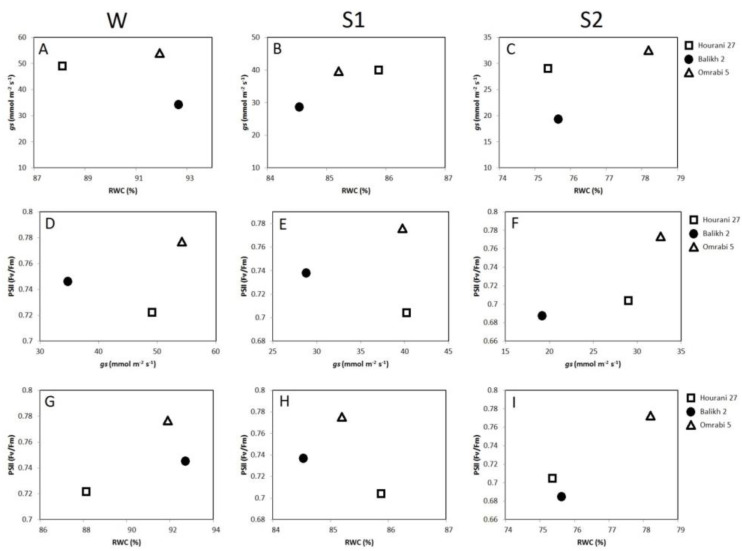
Relationship between pairs of physiological parameters in wheat for RWC vs. stomatal conductance (**A**–**C**), stomatal conductance vs. PSII activity (**D**–**F**) and RWC vs. PSII activity (**G**–**I**) in three durum wheat cultivars. Column panels (**A**,**D**,**G**), (**B**,**E**,**H**) and (**C**,**F**,**I**) represent W, S1 and S2, respectively.

**Figure 4 biotech-11-00043-f004:**
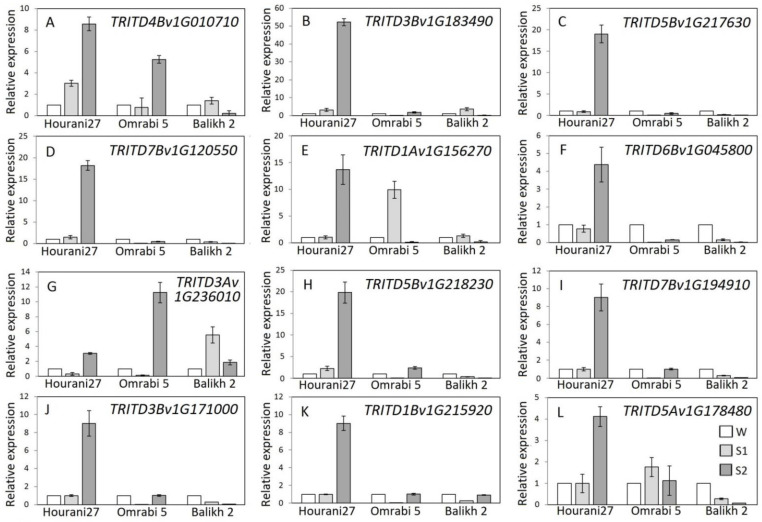
Relative expression of drought-responsive memory genes (indicated in the upper left corner of each panel) in three durum wheat cultivars (Hourani 27, Omrabi 5 and Balikh 2) during S1 and S2 stress stages as compared to the control W. Each subgraph (**A**–**L**) represent a different gene (gene name is located in the upper right corner of each subgraph). The actin 1 amplification was used as an internal reference gene. Data represent means ± 95% CI.

**Figure 5 biotech-11-00043-f005:**
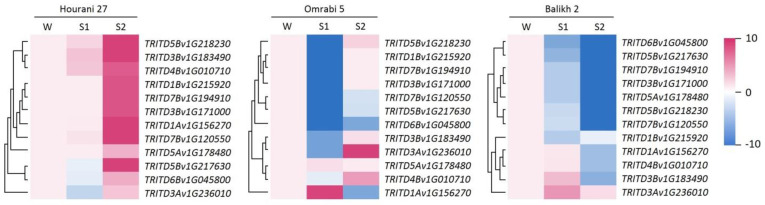
Heat maps of relative expression of drought-responsive memory genes in three durum wheat cultivars (Hourani 27, Omrabi 5 and Balikh 2) during S1 and S2 stress stages as compared to the control W. Presented folds range from −10 to 10.

**Figure 6 biotech-11-00043-f006:**
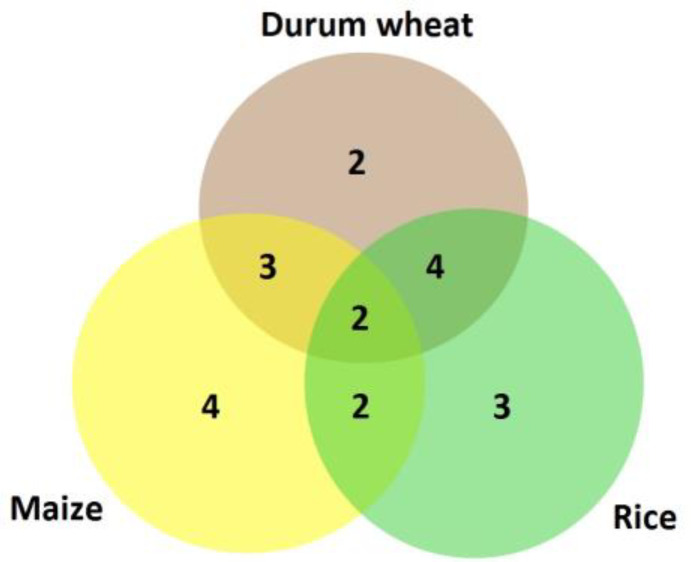
Venn diagram of similar expression response patterns during S1 and S2 stress stages for dehydration stress memory genes (a total of 11 genes) from durum wheat (*Triticum turgidum* L. ssp. *durum* (Desf.)) and orthologs from maize (*Zea mays*) and rice (*Oryza sataiva*).

**Figure 7 biotech-11-00043-f007:**
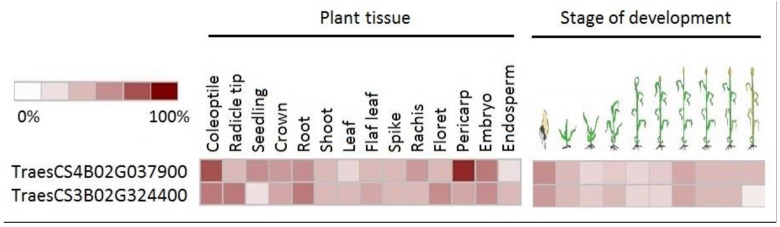
Transcript expression pattern for *TraesCS4B02G037900* (bread wheat homolog of durum wheat is *TRITD6Bv1G045800*, coding for Lipoxygenase 2) and *TraesCS3B02G324400* (bread wheat homolog of durum wheat is *TRITD3Bv1G171000*, coding for WRKY transcription factor). Expression is presented as a heat map in different plant tissues and during different stages of development during the lifecycle of bread wheat on publicly available expression data (www.genevestigator.com (accessed on 1 June 2022)).

**Table 1 biotech-11-00043-t001:** List of investigated dehydration stress memory genes in durum wheat (*Triticum turgidum* L. ssp. *durum* (Desf.)) and their description and primer sequences used to amplify them using qPCR, the last one corresponds to the reference gene actin.

Durum Locus Number	Gene Description	Forward and Reveres Primers 5′-3′
TRITD4Bv1G010710	Lipoxygenase 2	F-CTTCCATCGTCTACAAGAACTGG
		R-CCCGTCCACCGCGTACGGGTAGTC
TRITD3Bv1G183490	Protein kinase C-like zinc finger	F-GCGGAGCAAGTTCGCCTCCCAGACG
		R-GCCAGCCTCGCGGTGAACTTGACGC
TRITD5Bv1G217630	Basic helix-loop-helix (bHLH) DNA-binding	F-GTGCTGGTGCTGTTGCACAGCTGG
		R-CGATGTCCTCGTCCATCAGCTTCGC
TRITD7Bv1G120550	Transmembrane amino acid transporter	F-ATGTGGCTCATCATCTGCAAGCCC
		R-ATCTATGAGTAGAACTTGTATGTC
TRITD1Av1G156270	Late embryogenesis abundant (LEA)	F-CGTCCGAGACGGCCCAGGCCG
		R-GCTGTCTCCCCCCATCCCCAGC
TRITD6Bv1G045800	MYB transcription factor	F-AAGAGACCATGTTCAGAAGATAAC
		R-TCAGCATCTTCTTATCACACTGTTAC
TRITD3Av1G236010	Scarecrow-like protein (SCL1)	F-TCCAAGGGAAAGTCCAGATAGAATG
		R-GAATCCAGCCATCGTCATTCTCGCC
TRITD5Bv1G218230	Like/winged-helix DNA-binding family	F-GGAGACCAAGGCCAAGGCGGCCAAG
		R-GACGAACTTGGCGATGGCGTACGGG
TRITD7Bv1G194910	NAC transcription factor	F-CTAAGGGGAAGAAGACTGAGTGGG
		R-TCCCTGTGGGTAGCTTGGCAACGG
TRITD3Bv1G171000	WRKY transcription factor	F-GCGCAAGTACGGCCAGAAGCCCATC
		R-GTGATCGTAGGAGTAGGTGACGAGC
TRITD1Bv1G215920	Major facilitator superfamily	F-CGACGCTCGCCAACTGGCTGACTTC
		R-CCAAACTCATCTGTTGCACTTCCAC
TRITD5Av1G178480	AP2 transcription factor	F-CACGCAGTGTAAAGTTGTCGATAG
		R-GGAGCAGAGCAGTCCCAAAC
TRITD5Av1G093080	Actin	F-CCGAACGGGAAATTGTAAGG
		R-TCTCTGCCCCAATGGTGATC

**Table 2 biotech-11-00043-t002:** Expression response during S1 and S2 stress stages for dehydration stress memory genes from durum wheat (*Triticum turgidum* L. ssp. *durum* (Desf.)) and orthologs from maize (*Zea mays*) and rice (*Oryza sataiva*).

Durum Locus	Durum Response	Maize Locus	Maize Response *	Rice Locus	Rice Response **
TRITD4Bv1G010710	+/+, =/+, +/−	GRMZM2G102760	+/+	Os03g49380	+/−
TRITD3Bv1G183490	+/+, −/+, +/−	GRMZM2G106344	+/+	Os01g58194	+/+
TRITD5Bv1G217630	=/+, −/=	GRMZM2G004356	−/−	Os06g09370	+/+
TRITD7Bv1G120550	+/+, −/=	GRMZM2G429322	+/−	Os08g03350	+/−
TRITD1Av1G156270	=/+, +/−	GRMZM2G412436	+/−	Os02g15250	=/+
TRITD6Bv1G045800	=/+, −/=	GRMZM2G010920	−/+	OS03G25550	+/−
TRITD3Av1G236010	−/+, +/−	GRMZM2G153333	+/=	Os07g36170	−/+
TRITD5Bv1G218230	+/+, −/+, −/−	GRMZM2G401308	−/=	Os07g08710	−/+
TRITD7Bv1G194910	=/+, −/+, −/−	GRMZM2G063522	=/+	OS03G42630	+/−
TRITD3Bv1G171000	=/+, −/+, −/−	GRMZM2G013391	=/+	OS10G42850	=/+
TRITD1Bv1G215920	=/+, −/+	GRMZM2G028570	=/−	Os03g24870	+/−
TRITD5Av1G178480	=/+, +/−, −/−	GRMZM2G434203	=/−	Os04g46400	=/−

* Data from [19] and ** Data from [20].

## Data Availability

Not applicable.

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
