# Peer review of "Dehydration Stress Memory Genes in Triticum turgidum L. ssp. durum (Desf.)"

_biotech, 2022, doi:10.3390/biotech11030043_

Round 1

Reviewer 1 Report

The manuscript biotech-1891114, entitled “Dehydration stress memory genes in Triticum turgidum L. ssp. durum (Desf.)” reported and discussed the results of a laboratory experiment were the dehydration stress memory on durum wheat was assessed on several different genes. In particular, the authors compared the response of this genes expression on three different durum wheat varieties.   

In general, the manuscript and the experimental activity carried out seem to be of good quality following a strict scientific logic and according to widely used methods which have made it possible to obtain reliable results. In my opinion, only minor changes are needed before publication.

Introduction, it is of good quality, however I suggest to the authors to consider the follow manuscript (doi: 10.3389/fpls.2020.607226) that could help to improve it.

Materials and methods: are well detailed and do not needs revisions.

Results: if possible add more quantitative information in the text. Image quality must be improved.

Discussion: this part should be implemented considering a higher number of references/other studies. Anyway, its structure is fine:

Conclusions: must be rewritten summarizing the principal information given from this study.

My specific comments, which I hope will help the authors to improve their manuscript, are enclosed in the attached pdf file.

Author Response

Reviewer 1

Does the introduction provide sufficient background and include all relevant references?

Was improved as requested

Are the conclusions supported by the results?

Was improved as requested

The manuscript biotech-1891114, entitled “Dehydration stress memory genes in Triticum turgidum L. ssp. durum (Desf.)” reported and discussed the results of a laboratory experiment were the dehydration stress memory on durum wheat was assessed on several different genes. In particular, the authors compared the response of this genes expression on three different durum wheat varieties.

In general, the manuscript and the experimental activity carried out seem to be of good quality following a strict scientific logic and according to widely used methods which have made it possible to obtain reliable results.

Thank you for your positive comment.

In my opinion, only minor changes are needed before publication.

Thank you for all your valuable comments which were incorporated in the revised manuscript.

Introduction, it is of good quality, however I suggest to the authors to consider the follow manuscript (doi: 10.3389/fpls.2020.607226) that could help to improve it.

Thank you for your suggestion. The mentioned reference was added to the introduction.

Materials and methods: are well detailed and do not needs revisions.

Thank you for your positive comment.

Results: if possible add more quantitative information in the text.

Was added as requested

Image quality must be improved.

Was improved as requested

Discussion: this part should be implemented considering a higher number of references/other studies. Anyway, its structure is fine:

Additional references were added and used in discussion as requested 

Conclusions: must be rewritten summarizing the principal information given from this study.

Was improved as requested

My specific comments, which I hope will help the authors to improve their manuscript, are enclosed in the attached pdf file:

avoid to use in the keyword words that were mentioned in the title:

They were modified as requested

for this part of the introduction I suggest to consider the following manuscript doi: 10.3389/fpls.2020.607226

Thank you for your suggestion. The mentioned reference was added to the introduction.

for the future, I suggest to the authors to add also a not tolerant variety, as a control, in order to have a more complete analysis of phenomena.

Thank you for your suggestion that will be reflected in future studies.

… as earlier [26].

Was further clarified.

… R2 … also

Was corrected          

align the figures

were aligned as requested

… (P<0.05)…

Was Corrected

add a row-space

was added

Figure 4 please improve image quality

A higher resolution figure was used

conclusions must be rewritten summarizing the major results observed in the experiment.

Was improved as requested

Reviewer 2 Report

As you can see from my review, I was generally quite pleased with your work. Your presentation about gene expression by the three diverse durum cultivars was quite strong. On the other hand, I had a number of questions about the presentation of the physiological parameters. Please review the attached, annotated manuscript copy of comments. It is conceivable that I have misunderstood your figures and tables. If that is the case, my review may remain helpful, in that you will be prompted to more fully explain your affirmations (vis-a-vis, their statistical significance and practical importance).

Author Response

Reviewer  2

English language and style are fine/minor spell check required

Was improved as requested

Is the research design appropriate?

Was improved as requested

Are the methods adequately described?

Was improved as requested

Comments and Suggestions for Authors

As you can see from my review, I was generally quite pleased with your work.

Thank you for your positive comment.

Your presentation about gene expression by the three diverse durum cultivars was quite strong.

Thank you for your positive comment.

On the other hand, I had a number of questions about the presentation of the physiological parameters. Please review the attached, annotated manuscript copy of comments:

As a reviewer, I am very interested in these memory genes, triggered by repeated drought/dehyration stress cycles.

It is our pleasure to hear that

can you add "all" here?

Was added

". . . is required . . ." Also--are durums commonly grown in Jordan?

Was corrected. Yes, the majority of grown wheat varieties in Jordan are durum as they are more drought tolerant.

divers

Was corrected

You have efficiently and effectively introduced your subject, with supporting literature from both durum wheat and other crops (ones for which memory responses have been more fully characterized).

Thank you for your positive comment.

Good background on the cultivars studied.

Thank you for your positive comment.

previously described?

Was corrected

previously described?

Thank you for your positive comment.

So, when I look at Figure 2, it seems to me that differences in these cultivar responses are relatively modest (except for PSII for Omrabi 5 at S1. Or maybe I should say, except for Balikh 2 at S2 (looking at Figure 1). For my untrained eye, it looks like Figure one shows stepwise degradation of both RWC and gs--but that PSII seemed to be relatively insensitive to water stress cycles. I'm not clear why these two figures seem to be telling different "stories."

You are right. This is because figure (1) represent the absolute mean values, which is important to illustrate the change in physiological parameters for each cultivar. On the other hand, figure (2) represents “RELATIVE” values compared to the control (W), this is helpful to illustrate the percentage change in each parameter compared to the initial value (control) after each stress event.

Figure 3 (and its associated text) seems to display ratios between the parameters of interest. Not clear to me if these ratios were subjected to statistical testing. Further, it seems like these differences may be relatively modest. Can you establish that these cultivar differences are both significant and of practical importance?

You are right again. This type of relationship plots is another way of representing any cluster of plotted data, Please see “Shavrukov, Y., Langridge, P., Tester, M., & Nevo, E. (2010). Wide genetic diversity of salinity tolerance, sodium exclusion and growth in wild emmer wheat, Triticum dicoccoides. Breeding science60(4), 426-435.” It is similar to PCA plot but rather with limited plotting data. The significant differences between measurements are presented in figure (1)

I hope that you explain to your readers what you think it means that durum's responses were so diverse across cultivars, as compared with maize and rice.

This was explained as requested in discussion section.

As far as gene expression goes, your data seem to show strong and significant differences (to a much greater degree than the above physiological measurements).

This was explained as requested in conclusion section.

A key inference about the source of the physiological memory--dependent upon a transcriptional memory.

Yes that was also evident here

Another strong inference, based on your data.

Thank you for your comment.

It is conceivable that I have misunderstood your figures and tables. If that is the case, my review may remain helpful, in that you will be prompted to more fully explain your affirmations (vis-a-vis, their statistical significance and practical importance).

Not at all, we highly appreciate all your comments. I have explained your concerns above and in the revised manuscript